# Immune-Related Adverse Events in a Patient Treated with Pembrolizumab: A Case Report from the Point of View of a Geriatrician

**DOI:** 10.3390/geriatrics9060160

**Published:** 2024-12-11

**Authors:** Philipp Oft, Markus Gosch, Francesco Pollari

**Affiliations:** 1Medizinische Klinik 2–Geriatrie, Klinikum Nürnberg-Paracelsus Medical University, 90419 Nuremberg, Germany; markus.gosch@klinikum-nuernberg.de; 2Klinik für Herzchirurgie, Klinikum Nürnberg-Paracelsus Medical University, 90471 Nuremberg, Germany; francesco.pollari@klinikum-nuernberg.de

**Keywords:** geriatrics, immune-related adverse events, pembrolizumab, immune myocarditis, third-degree atrioventricular block, immune checkpoint inhibitor

## Abstract

We report the case of a 78-year-old female patient who received palliative immunotherapy with pembrolizumab and lenvatinib as a treatment of pulmonary and osseous metastatic endometrial carcinoma. Under this therapy, the patient developed dysphagia, thyroiditis with hypothyroidism, myositis, and myocarditis, which required, due to third-degree AV block, the installation of a pacemaker. The patient received high-dose cortisone therapy, a thyroid hormone substitution, and pyridostigmine for symptom control. With this therapy, we saw a significant but not complete regression of symptoms. Ultimately, we could discharge the patient home for an outpatient treatment. The case report is followed by a discussion of the management of immune-related adverse events (irAEs) during pembrolizumab therapy from a geriatric perspective. Elderly patients on pembrolizumab therapy require close monitoring for irAEs, which can present atypically or without symptoms and may be fatal. Non-invasive diagnostics and minimizing hospital stays are essential to preserve the fitness of this vulnerable population.

## 1. Introduction

Pembrolizumab is a monoclonal antibody that has been approved in combination with lenvatinib for the treatment of advanced endometrial carcinoma since 2019 [1]. Patients receive an intravenous application of 200 mg every third week for up to 24 months [2]. Pembrolizumab exerts its effects via the inhibition of PD-1. PD-1 is an important factor in the body’s immune response and its fight against tumor cells. Some carcinomas, e.g., breast cancer, melanoma, or non-small-cell lung cancer [3], express the ligands PD-L1 and PD-L2, which decrease T-cell activity by binding to the PD-1 receptor [4]. PD-1 inhibition therefore results in the reactivation and proliferation of cytotoxic T-cells. However, this process can also be accompanied by so-called immune-related adverse events (irAEs). These are generally classified into five severity levels [5]. A total of 18.50% of all patients experience an irAE during pembrolizumab therapy and 5.10% experience a severe one [6]. The most common side effects are fatigue, pruritus, skin rash, and diarrhea [4].

## 2. Case Presentation

This report presents the case of a 78-year-old female patient with osseous and pulmonary metastatic endometrial carcinoma (initial diagnosis (ID), 22 years before; Figure 1). The only known pre-existing conditions were a cataract, which was operated on a few months before admission to the hospital, and a middle cerebral artery stroke in the previous year. The main symptoms of the stroke were dysarthria, as well as a left-sided leg and facial hemiparesis, both of which completely regressed within 24 h. The risk stratification only showed hypercholesterolemia and small atherosclerotic plaques in both carotid bifurcations without hemodynamic relevance.

Therefore, the premedication consisted of acetylsalicylic acid at 100 mg and atorvastatin at 20 mg only. A hysterectomy had already been performed at the time of the ID. The first pulmonary and osseous metastases were discovered 7 years before and a therapy was started one year later by radiation and a combined therapy of progestin and bisphosphonates. There was a switch to tamoxifen in 2019 and in 2022 to palliative chemotherapy with carboplatin and paclitaxel. Finally, in May 2023, a therapy with pembrolizumab and lenvatinib was started. At the time of admission, the patient had already received three cycles of pembrolizumab, with the last one administered 12 days prior to hospitalization. The reason for hospitalization was a ptosis—more pronounced on the left side than on the right side—as well as a new onset of dysphagia, speech disorders, and a loss of muscle strength, which made it particularly difficult to keep the head upright. All symptoms occurred simultaneously approximately 10 days before admission. The patient did not develop a fever at any time. As mentioned above, the patient was not on any cardiovascular drugs. The admission ECG showed a third-degree AVB alternating with a second-degree Mobitz II type at heart rates between 35 and 55 bpm (Figure 2A). A previous ECG performed 5 months before was available and showed a regular sinus rhythm with an indifference type and a regular R wave transition from V3 to V4 (Figure 2B).

The patient herself did not report any corresponding cardiac symptoms such as dizziness, syncope, or weakness, even when explicitly asked. Two days after admission, a two-chamber permanent pacemaker was implanted without complications. Laboratory diagnostics on admission revealed latent hypothyroidism with elevated thyroid-stimulating hormone (TSH = 12.40 μ IU/mL) and thyroid hormones in the normal range (T3 = 1.95 pg/mL; T4 = 1.07 ng/dL). A still regular TSH (0.49 μ IU/mL) was measured during routine follow-up under pembrolizumab therapy two weeks before admission (Figure 3C).

Infection parameters were within normal range, and renal function was only mildly impaired with a creatinine level of 0.96 mg/dL and an estimated GFR of 55.9 mL/min. Cardiac enzymes (Figure 3A,B) were markedly elevated (CK = 910 U/l, CK-MB = 81.50 ng/mL, cTnT = 1.24 ng/mL). The antibodies taken (anti-AchR-AK, anti-SOX1-AK, anti-LRP4-AK, anti-VGCC-AK) were all negative. Transthoracic echocardiography performed one day after pacemaker implantation showed moderately decreased systolic left ventricular function with a left ventricular ejection fraction of 41%, with apical akinesia, as well as apicalseptal, apical inferior, and anterior hypokinesia. Additional findings included mild mitral regurgitation and moderate tricuspid regurgitation. A transthoracic echocardiography, which was performed for risk stratification after the stroke one year before, showed a good left ventricular function without pathologies. We interpreted the result of the echocardiography in conjunction with elevated cardiac enzymes and the ECG changes as immune myocarditis. According to the definition of the European Society of Cardiology, the reported case technically displays a *clinically suspected myocarditis*, as we did not perform heart MRI or biopsy [7]. The electromyography showed positive sharp waves as well as low-amplitude motor units with shortened polyphasic potentials. This finding, in conjunction with the markedly elevated creatine kinase (Figure 3A) and typical clinic, confirmed the suspicion of myositis. However, the examined muscles showed no pathological decrement in serial stimulation, so there was no evidence of myasthenic exhaustibility. Nevertheless, a pyridostigmine test was performed, under which the patient described a subjective improvement in dysphagia and muscle weakness and an improvement in ptosis was observed. In addition, a barium swallow examination was initiated, in which a regular swallow act was observed. An MRI of the head, which was performed in the outpatient setting one week before admission due to the symptoms, also showed no pathology. The computer tomography scans of the neck and thorax did not show any new lesions except for the known filiae, which were slightly regressed compared to the previous examination. For further clarification, a cardiac MRI would also have been useful but was not possible due to the recent pacemaker implantation. Once the diagnostic process was completed, Myasthenia gravis, Lambert Eaton syndrome, and Horner syndrome could be excluded as possible differential diagnoses and an irAE was confirmed (Figure 1). We started cortisone therapy with 100 mg prednisolone i.v. for 5 days at a body weight of 53.2 kg, which was subsequently phased out orally over 6 weeks. In addition, despite the absence of decrement with a subjective improvement of symptoms, the patient initially received 30 mg pyridostigmine four times daily—with the option of increase if well tolerated—and 25 μg L-thyroxine by the manifested hypothyroidism (TSH: 45.80 micro IU/mL, T3: 1.56 pg/mL, T4: 0.5 ng/dL; see Figure 3C), which should be considered as another irAE most likely in the context of thyroiditis. During her stay, the patient received supportive physiotherapy, occupational therapy, and logopedic therapy. Overall, a significant improvement in dysphagia and muscle strength was observed during this therapy. Furthermore, the Barthel motor index [8] improved from 70 to 85 (maximum of 100 points) during the stay, and the timed-up-and-go test [9] decreased from 30 to 21 s. With therapy, cardiac and muscle enzymes additionally showed a marked decrease (Figure 3A,B). Hypothyroidism persisted until the time of discharge, as we were careful to compensate slowly. Clinically, we saw a marked improvement in ptosis (Figure 4) and previously existing speech impairments. One week after the start of therapy (and 16 days after hospital admission), the patient could be discharged home (Figure 1). Immunotherapy was not resumed, as no tumor progression was detected in the subsequent examinations. The patient’s wish to fully discontinue tumor therapy was prioritized to preserve the patient’s autonomy and minimize the impact on her quality of life. 

## 3. Discussion

Although there are previous case reports of patients with myocarditis and associated third-degree AVB [10], to our knowledge, this is the first report of a patient additionally developing myositis and thyroiditis during pembrolizumab therapy. However, hypothyroidism has also been described in lenvatinib monotherapy, but the incidence is markedly increased in combination with pembrolizumab [11]. Importantly, the focus here will be on the events from a geriatric point of view, encompassing not only frail or severely impaired patients but also those starting treatment in a fit condition, as even these individuals may face unique challenges during ICI therapy. We applied the principle of keeping the hospital stay as long as necessary and as short as possible. The aim was an early discharge after completed diagnostics and first recognizable therapy successes, as well as further outpatient control of the patient. In our view, there are several reasons for this. First, it has been shown that a prolonged hospital stay leads to an increased risk of nosocomial infections, especially in geriatric patients [12]. In addition, during hospitalization, patients engage in much less physical activity compared to their routine at home. This is particularly the case for elderly patients [13]. Associated with such increased immobility is a marked decrease in lower extremity strength [14]. A significant deterioration in sleep quality has been observed in geriatric patients, as well as an associated increase in blood pressure [15]. Likewise, we try—for each of our geriatric patients—to keep diagnostics low and as non-invasive as possible. In a report, Schiopu et al. [16] presented the case of a 75-year-old male patient who was treated with pembrolizumab for malignant mesothelioma and suffered from similar symptoms. In this case, the authors performed a cardiac catheter examination, which did not show any coronary abnormalities. We therefore decided against cardiac catheterization in regard of the risks, such as vascular injuries or infections [17], even if the probabilities were low. This is equally relevant for muscle or myocardial biopsies. In addition to blood tests, we used well-known geriatric assessments such as the Barthel motor index [8] and the timed-up-and-go test [9] to monitor the success of therapy. Geriatric assessments are valuable tools for evaluating therapeutic outcomes in elderly patients. While not consistently implemented in geriatric case reports [18,19], their application can provide a more nuanced understanding of patient progress and treatment impact. In comparison to other case reports [16,20], we decided to use an initial low dose of prednisolone—about 2 mg/kg of body weight—to minimize the side effects of glucocorticoid therapy such as infections [21], hypertension, or osteoporosis [22]. To evaluate the success of the therapy, geriatric assessments were performed from the start of hospital admission. To adequately monitor patients at risk and select those who are likely to benefit from immune checkpoint inhibition (ICI) therapy, in a geriatric setting, it is necessary to assess the risks and efficiency. Therefore, we want to take a closer look at our patient’s risk profile. As described above, there are no other relevant pre-existing conditions and medications, so relevant factors are age and gender. With increasing age, the efficacy of immunotherapy decreases; accordingly, adverse events are less likely in older patients (>75 years) [23]. However, age-related research on ICI efficiency is rather limited. Generally, responses to immunotherapy are heterogeneous. While differences have been observed in CTLA-4 inhibitors, such differences are not clearly evident in PD-1 inhibitors [24]. Interestingly, very elderly patients (>85 years) do not appear to have a disadvantage compared to patients aged 80–85 years when treated with single-agent immune checkpoint inhibitors [25]. Studies report varying results regarding the susceptibility to and severity of immune-related adverse events in relation to age. However, the data suggest that younger patients may have a higher risk of developing severe irAEs (median age 63.5 to 66.7 years) [26]. Conversely, the risk of fatal adverse events appears to be higher in older patients (median age 70 years vs. 62 years) [27]. In elderly patients, dermatologic and rheumatologic irAEs occur more frequently, while endocrine and gastrointestinal irAEs are commonly observed in younger patients [28]. Of note, in a meta-analysis including 103 randomized studies conducted by Zhong et al. [29], males showed improved overall survival (OS) with ICI treatment over females, most pronounced in PD-1-targeting therapies. The authors suggest that heterogeneity in general immune response intensity and checkpoint molecule expression are key factors in those differences. Overall, the risks for irAEs do not differ significantly between men and women. However, they vary depending on the type of irAE and the specific immune checkpoint inhibitor used [30]. To summarize, our patient had an increased risk of fatal side effects due to her gender and age and a lower chance to benefit from the therapy. It is therefore important to weigh the risk against the benefit before starting therapy and to ensure adequate monitoring of the side effects. Finally, we would like to focus on the course of the initial symptoms and the general monitoring of pembrolizumab therapy. In our case report, the initial hospitalization was only due to a new-onset ptosis of the left eye and the patient’s dysphagic complaints. She did not report cardiac symptoms at any time, such as dyspnea, fatigue, or syncope. If a routine ECG had not been performed on admission, the myocarditis would not have been prominent initially. This could have led to fatal consequences for the patient. Although the cardiac dangers of PD-1 inhibitor therapy have been reported in several case reports [10], an ECG was not performed on our patient before starting therapy or between pembrolizumab administrations. Therefore, as an inexpensive and noninvasive rapid method, regular ECG checks should be considered for future therapy monitoring.

## 4. Conclusions

In this report, we reviewed the case of a series of irAEs under pembrolizumab therapy. We demonstrate that elderly patients receiving ICI therapy require heightened vigilance for irAEs. The number of studies focusing on geriatric patients currently seems to be insufficient. Further studies, preferably prospective, would be desirable to balance thorough evaluation with patient well-being. Symptomless but potentially lethal irAEs must be considered, as they might initially be clinically silent, as shown in our case. For this purpose, a rapid, noninvasive general basic diagnostic, which always needs to include a carefully performed physical examination, cardiac monitoring, and the prompt initiation of glucocorticoid therapy, is recommended for geriatric patients. In stable patients and those with recognizable therapeutic success, prompt discharge and further outpatient care is possible and preferable to minimize long-term impacts on patient fitness.

## Figures and Tables

**Figure 1 geriatrics-09-00160-f001:**
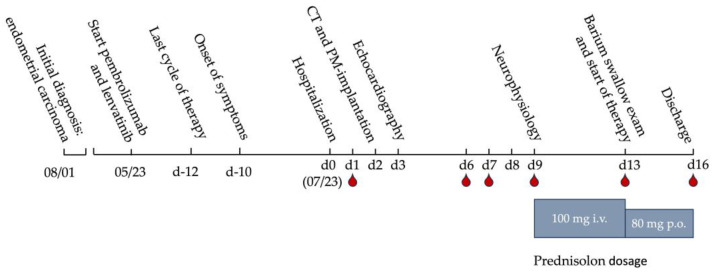
Overview of the time sequence before hospitalization, the performed diagnostics, and important events during hospitalization; blood draws are marked with a red drop below the timeline; additionally marked in blue are the doses of prednisolone therapy. CT: computer tomography; PM: pacemaker.

**Figure 2 geriatrics-09-00160-f002:**
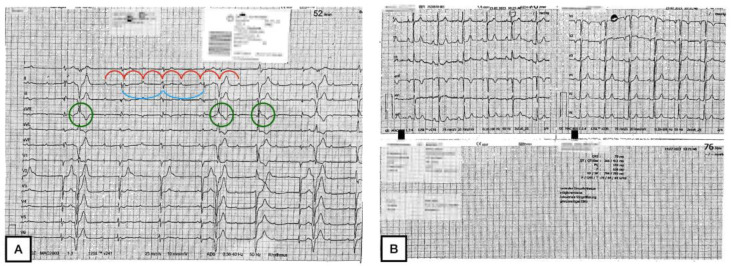
Changes in Electrocardiography. (**A**) The 12-lead ECG at admission: third-degree AV block with a heart rate of 52 bpm and over-rotated left type; The atrial rhythm with a regular P-to-P interval; marked in red. The escape rhythm with a regular R-to-R interval; marked in blue, which is interrupted by numerous ventricular extrasystoles; marked in green. (**B**) The 12-lead ECG dated 5 months before: a regular sinus rhythm with a heart rate of 76 bpm; indifference type; regular R wave transition between V 3 and V 4.

**Figure 3 geriatrics-09-00160-f003:**
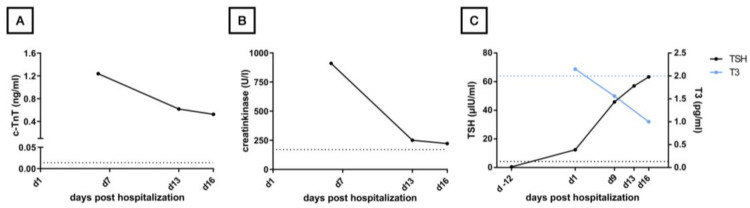
The time course of measured laboratory parameters. Day 0 is defined as the day of hospital admission; day 16 as the day of discharge; the start of therapy on day 9; dashed lines each mark the border of the reference range (RR). (**A**) cTnT as a parameter for the course of myocarditis after the start of therapy (RR = 0.014 ng/mL). (**B**) Creatine kinase as a course parameter for myositis (RR = 170 U/l). (**C**) The time course of thyroiditis and associated hypothyroidism with TSH (black, left scale, RR = 4.2 μIU/mL) and T3 (blue, right scale, RR = 2.0 pg/mL). cTnT, cardiac troponin T; RR, reference range; TSH, thyroid-stimulating hormone; T3, triiodothyronine.

**Figure 4 geriatrics-09-00160-f004:**
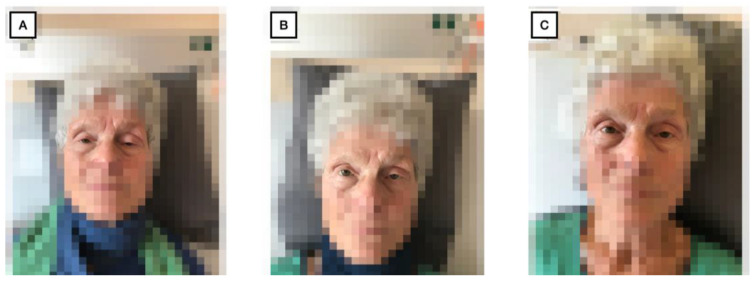
Clinical course of ptosis under glucocorticoid therapy; on the day of initiation of therapy (**A**), the day after initiation of therapy (**B**), and 3 days after initiation of therapy (**C**). Reproduced with permission of the patient.

## Data Availability

The original contributions presented in this study are included in the article. Further inquiries can be directed to the corresponding author(s).

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
