# Peer review of "Immune-Related Adverse Events in a Patient Treated with Pembrolizumab: A Case Report from the Point of View of a Geriatrician"

_geriatrics, 2024, doi:10.3390/geriatrics9060160_

Round 1

Reviewer 1 Report

Comments and Suggestions for Authors

The case report is well presented and accurate. The message by the authors to pay particular attention to geriatric patients (under Pembrolizumab therapy) is clear and worth consideration as well as the suggestion of performing regular ECG checks. However, as this is not the first case of the mentioned irAEs, I suggest the authors discuss these cases in the discussion section further highlighting the novelty of this case. In addition, I think that a discussion of the results on gender differences might be relevant. I have a major ethical concern regarding the publication of patients’ photos: It is unlikely that informed consent asked explicitly for the photos consent, so I ask the authors to verify this aspect and, if necessary, to censor the face.

Author Response

Comment 1:

The case report is well presented and accurate. The message by the authors to pay particular attention to geriatric patients (under Pembrolizumab therapy) is clear and worth consideration as well as the suggestion of performing regular ECG checks. However, as this is not the first case of the mentioned irAEs, I suggest the authors discuss these cases in the discussion section further highlighting the novelty of this case.

Response 1:

Thank you for your valuable suggestion. We have expanded the discussion of previously described cases of irAEs in the discussion section, ensuring that these are appropriately acknowledged (lines 167 - 171, 176 - 178, 178 - 181). In addition to the initially mentioned novelty of our case with its particular geriatric focus, this further addresses the uniqueness of our case by embedding it within the context of existing literature. We hope that these revisions meet your expectations.

Comment 2:

In addition, I think that a discussion of the results on gender differences might be relevant.

Response 2:

Thank you for your helpful suggestion. In response, we have added a section discussing gender differences (lines 200 – 206), specifically focusing on the variations in the efficacy of ICI, as well as the likelihood of irAEs. We believe this addition provides a more comprehensive view of the topic and addresses the gender relevance.

Comment 3:

I have a major ethical concern regarding the publication of patients’ photos: It is unlikely that informed consent asked explicitly for the photos consent, so I ask the authors to verify this aspect and, if necessary, to censor the face.

Response 3:

Thank you for expressing your concern regarding the patient’s privacy. The declaration of consent signed by the patient does include permission to use her photo; however, to further address this matter, we additionally censored the patient’s face (Fig. 4).

Reviewer 2 Report

Comments and Suggestions for Authors

The title of the manuscript is: „Immune-related adverse events in a patient treated with Pembrolizumab - A case report from a point of view of a geriatrician”. It is an interesting case report about the immune-related side effects of antiPD1 therapy in an endometrial cancer patient in advanced age. Over the specialties and values of the story, I have several comments/notifications about it.

1, I have missed the oncological point of view considering the whole manuscript. The reader has no information about the future treatment plan of the patient, the final discontinuation of pembrolizumab, there was no discussion about the potential role of lenvatinib in the thyroid dysfunction etc. I recommend some dedicated and separate discussion about special IrAEs of antiPD1 treatments in the elderly ages.

2, One important criterion for starting a combination of immunotherapy and targeted agents is the acceptable performance state (ECOG 0-2) of the patients. Based on this argument the general geriatric concerns (see from line 142) are not valid for a cancer patient receiving combined modalities treatment. Please refine this part of the Discussion section.

3, The authors emphasize the necessity of prompt initiation of steroid medication in case of IrAE (see line 174). However, there was 6 days interval between the first diagnosis and the introduction of steroid therapy (see Figure 4.). I realized that the treating physicians initiated other exams as well, but was there any differential diagnostic debate? Please clarify it.

4, Minor concerns: What is ID: initial diagnosis (see line 37)? Please clarify it.

5, Finally, I suggest some general message (special symptoms, increased attention etc.) to the readers (in Abstract and/or in Conclusion sections) considering immune therapy in the elderly ages.

Comments on the Quality of English Language

I am not a native English speaking one, but I recommend some medical English overview of the text.

Author Response

General response:

Thank you for your valuable feedback and for highlighting the key aspects for improvement. As recommended, we have revised the language of the case report to enhance clarity and precision.

Comment 1:

I have missed the oncological point of view considering the whole manuscript. The reader has no information about the future treatment plan of the patient, the final discontinuation of pembrolizumab, there was no discussion about the potential role of lenvatinib in the thyroid dysfunction etc. I recommend some dedicated and separate discussion about special IrAEs of antiPD1 treatments in the elderly ages.

Response 1:

Thank you for your helpful feedback. We appreciate your insights regarding the oncologist’s perspective. While we have maintained the primary focus of our manuscript on the geriatric viewpoint, we have incorporated the oncological aspects you highlighted. Specifically, we have included a remark on the discontinuation of Pembrolizumab therapy (lines 138 - 141). Additionally, we have addressed the potential role of Lenvatinib (lines 151 - 153) and added a discussion on irAE in elderly patients (lines 194 - 200).

Comment 2:

One important criterion for starting a combination of immunotherapy and targeted agents is the acceptable performance state (ECOG 0-2) of the patients. Based on this argument the general geriatric concerns (see from line 142) are not valid for a cancer patient receiving combined modalities treatment. Please refine this part of the Discussion section. 

Response 2:

Thank you for your insightful comment. We understand your point regarding the use of EGOG 0-2 as a criterion for combination therapies. However, we believe that general geriatric concerns remain relevant for elderly patients, even those who are fit and meet the criteria for combined treatment. As highlighted in our manuscript, elderly patients, even without significant pre-existing functional limitations, are still at risk of functional decline during hospitalization due to factors such as restricted mobility and increased vulnerability to infections. This decline, coupled with reduced regenerative capacity, underscores the need for a comprehensive geriatric approach. We have added a clarification in the discussion section of the manuscript to emphasize that these concerns generally apply to geriatric patients, including those who are otherwise fit and eligible for combined therapies (lines 153 - 156). We hope this addition helps to address your concern while maintaining the focus on the importance of the geriatric perspective.

Comment 3:

The authors emphasize the necessity of prompt initiation of steroid medication in case of IrAE (see line 174). However, there was 6 days interval between the first diagnosis and the introduction of steroid therapy (see Figure 4.). I realized that the treating physicians initiated other exams as well, but was there any differential diagnostic debate? Please clarify it. 

Response 3:

Thank you for your comment. The delay in initiating steroid therapy was due to the need to complete diagnostic examinations to rule out differential diagnosis. Additionally, the placement of the pacemaker was prioritized at the time. We have revised the section discussing the differential diagnostic process to clarify this point (lines 119 - 122).

Comment 4:

Minor concerns: What is ID: initial diagnosis (see line 37)? Please clarify it. 

Response 4:

Thank you for addressing this omission. We have added the clarification of the abbreviation (line 41).

Comment 5:

Finally, I suggest some general message (special symptoms, increased attention etc.) to the readers (in Abstract and/or in Conclusion sections) considering immune therapy in the elderly ages.

Response 5:

We appreciate your suggestion. We have added a general message to the abstract (lines 19 - 22) and revised the conclusion section (line 222 - 224, 229 - 231) highlighting the key considerations for managing elderly patients undergoing immune checkpoint inhibition. We hope this addition addresses your recommendation and provides valuable guidance for the reader.

Reviewer 3 Report

Comments and Suggestions for Authors

This case report describes a 78-year-old female patient with pulmonary and osseous metastatic endometrial carcinoma who underwent palliative immunotherapy with pembrolizumab and lenvatinib. The patient developed dysphagia, thyroiditis with hypothyroidism, myositis, and myocarditis during treatment, necessitating a pacemaker implantation. Despite symptom improvement with therapy, complete regression was not achieved, and the patient was discharged for outpatient treatment. However, the following issues are required for explaining:

1. The authors should clearly outline the timeline of the patient's use of immunotherapy and symptom occurrence in Figure 4, considering moving this information to Figure 1 for better clarity.

2. The electrocardiogram images need to be of higher resolution with specific abnormalities marked using arrows.

3. What age threshold does the author consider as "elderly" for assessing the efficacy of immunotherapy in tumor patients - 60 years or 65 years?

4. It would be beneficial to provide clinical evidence on the increased risk of immune-related adverse events in elderly tumor patients undergoing immunotherapy.

5. How can the experiences shared in this case report be applied in future clinical practice, emphasizing key discussion points for better understanding and implementation?

Comments on the Quality of English Language

The authors are recommended to consider engaging a professional language editing service to ensure the clarity and coherence of the manuscript.

Author Response

General response:

We appreciate your insightful feedback. As recommended, we have revised the language of the case report to enhance clarity and precision.

Comment 1:

The authors should clearly outline the timeline of the patient's use of immunotherapy and symptom occurrence in Figure 4, considering moving this information to Figure 1 for better clarity.

Response 1:

Thank you for this helpful suggestion. We have revised the figure and added the respective information, we have further moved it ahead to improve clarity (Fig. 1).

Comment 2:

The electrocardiogram images need to be of higher resolution with specific abnormalities marked using arrows.

Response 2:

We appreciate your helpful comment. We have provided the electrocardiogram images in higher resolution and added marks to highlight abnormalities (Fig. 2).

Comment 3:

What age threshold does the author consider as "elderly" for assessing the efficacy of immunotherapy in tumor patients - 60 years or 65 years?

Response 3:

Thank you for stating your concern. We have added the age referred to for the studies mentioned in this case report for clarification (lines 188, 192, 193, 196, 198).

Comment 4:

It would be beneficial to provide clinical evidence on the increased risk of immune-related adverse events in elderly tumor patients undergoing immunotherapy.

Response 4:

We appreciate this insightful suggestion. We have added a section discussing the risk and severity of immune related adverse events in elderly tumor patients treated with immune checkpoint inhibition (lines 194 - 200).

Comment 5:

How can the experiences shared in this case report be applied in future clinical practice, emphasizing key discussion points for better understanding and implementation?

Response 5:

Thank you for this helpful comment. We have added a section emphasizing the key messages to the abstract (line 19 - 22) and revised the conclusion section (lines 222 - 224, 229 - 231) highlighting the key considerations for future application in clinical practice. We hope this addition improves understanding, enables implementation and provides useful guidance for the reader.

Round 2

Reviewer 1 Report

Comments and Suggestions for Authors

The authors have addressed all my previous comments. 

Reviewer 3 Report

Comments and Suggestions for Authors

The revised manuscript has made a great improvement. I have no more comments and recommends.